# 17βH-Neriifolin Improves Cardiac Remodeling Through Modulation of Calcium Handling Proteins in the Heart Failure Rat Model

**DOI:** 10.3390/biomedicines13092115

**Published:** 2025-08-29

**Authors:** Rajasegar Anamalley, Yusof Kamisah, Nurhanan Murni Yunos, Satirah Zainalabidin

**Affiliations:** 1Programme of Biomedical Science, Centre of Toxicology and Health Risk Study (CORE), Faculty of Health Sciences, Universiti Kebangsaan Malaysia, Kuala Lumpur 50300, Malaysia; rajasegar@msu.edu.my; 2Faculty of Health and Life Sciences, Management and Science University, University Drive, Shah Alam 40100, Malaysia; 3Department of Pharmacology, Faculty of Medicine, Universiti Kebangsaan Malaysia, Kuala Lumpur 56000, Malaysia; kamisah_y@hctm.ukm.edu.my; 4Cardiovascular and Pulmonary Research Group, Universiti Kebangsaan Malaysia, Bangi 43600, Malaysia; 5Natural Products Division, Forest Research Institute Malaysia (FRIM), Kepong 52109, Malaysia; hanan@frim.gov.my

**Keywords:** cardiac glycoside, isoprenaline, cardiac hypertrophy, heart failure

## Abstract

**Background**: Cardiac glycosides such as digoxin have been commonly used for patients with heart failure; however, their toxicity remains a main concern. 17βH-neriifolin (SNA209), a cardiac glycoside compound, has been recently isolated from Ceberra odollum Gaertn and was shown to improve the heart’s pumping ability in failing hearts ex vivo. Thus, this study aimed to investigate the potential use of SNA209 as a treatment for isoprenaline (ISO)-induced heart failure in rats. **Methods**: Forty male Wistar rats were randomly divided into five groups. Heart failure was induced by isoprenaline (ISO, 10 mg/kg/s.c) for 14 days daily, followed by SNA209 treatment (5 mg/kg; p.o) for another 14 days daily. Control rats were given saline as a vehicle for ISO and DMSO as a vehicle for SNA209. **Results**: Systolic and diastolic blood pressure (SBP and DBP) in all ISO-treated groups were significantly increased compared to the control group (*p* < 0.05), and SNA209 treatment managed to reduce the SBP and DBP. Additionally, SNA209 treatment significantly increased the heart rate and normalized the ECG parameters in ISO-treated rats. Pro-B-type natriuretic peptide and troponin T level, a cardiac injury markers, was remarkably reduced by SNA209 in the ISO-treated group. Cardiac hypertrophy was evident in increased cardiomyocyte size in ISO groups; however, SNA reduced the cardiomyocyte size. The left ventricular developed pressure (LVDP) in ISO treated with SNA209 was significantly raised, indicating a chronotropic effect. Cardiac Na^+^/K^+^-ATPase expression of the α1 subunit, sarcoplasmic/endoplasmic reticulum Ca^2+^ ATPase 2a (SERCA2a), and sodium–calcium exchanger subunit were significantly increased in the SNA treatment groups. **Conclusions**: The SNA 209 treatment improved cardiac function and structure, likely via modulating intracellular calcium management, so underscoring its potential as an adjuvant therapy for heart failure.

## 1. Introduction

In the last ten years, there has been significant progress in our understanding and management of heart failure. However, as life expectancy rises and other illnesses that might cause heart failure emerge, the disease’s incidence, prevalence, mortality, and monetary costs continue to rise. According to the Global Burden of Disease 2020, heart failure remained at a high prevalence and a major risk of death among older persons from 1990 to 2020 [1]. In heart failure patients, anomalies of cardiac structure and function are detected, such as a reduction in ventricular filling and/or ejection capacity [2]. Initially, one of the first drugs used to resolve this heart condition issue was digoxin, a cardiac glycoside derived from the plant *Digitalis lanata*. It has been widely used to treat a variety of heart conditions, such as congestive heart failure, atrial fibrillation or flutter, and certain cardiac arrhythmias. However, the toxicity remains a significant concern because of its adverse effects. The incidence of side effects is estimated at up to 20% overall; amongst 50% are cardiac symptoms, 25% are gastrointestinal tract symptoms, and the remainder are manifestations of CNS and other side effects [3]. As such, although conventional treatment helps treat cardiovascular problems, there are still significant obstacles to overcome, including expensive lifelong management and the rebound phenomenon. An alternative form of treatment for heart failure uses a variety of plants and active ingredients with few adverse effects.

Recently, a novel cardiac glycoside compound identified as 17βH-neriifolin (SNA209) from *Cerbera odollam* leaves has been isolated. *Cerbera odollam*, also known as the suicide tree or Pong Pong, is a flowering plant native to India and certain parts of Southern Asia. *C. odollam* contains cerberin, a cardiac glycoside that inhibits the Na^+^/K^+^ATPase pumps and calcium channels in the heart muscle, resulting in deadly arrhythmias [4]. The early production of cardiac glycosides in *C. odollam* is assumed to be a secondary metabolism for chemical defenses against herbivores and insects. An evolutionary conserved cardiac glycoside binding site in the Na^+^/K^+^ATPase is present in a variety of species, including insects, herbivores, and humans. For these reasons, extracts from *C. odollam* are used to make pesticides to control pests [5]. The anticancer potential of cardiac glycosides extracted from *C. odollam* was also mentioned in a few past studies. Data from in silico and in vitro malachite green assay show that 17βH neriifolin (SNA209) binds to the α-subunit pocket of Na^+^/K^+^-ATPase [6]. Nevertheless, it is still unknown whether 17βH neriifolin, isolated from *C. odollam*, has any cardiac inotropic effects. Hence, this study aimed to evaluate the effects of 17βH-neriifolin on cardiac protein expression, cardiac structure, and function in the heart failure rat model.

## 2. Materials and Methods

### 2.1. Test Compound Isolation

17βH-neriifolin (SNA 209) was isolated from the leaves of *C.odollam* based on the previously described method in Siti Syarifah et al. [7]. *C. odollam* plant species were collected at Kuala Selangor, Selangor and authenticated by a botanist and deposited in the herbarium of the Research Institute Malaysia, Kepong, Selangor, Malaysia with voucher specimen number of AC5832-P.

### 2.2. Animal

Forty male Wistar rats (200–250 g) were used in this study. The rats were acclimatized under standard laboratory conditions for 1 week with a 12-hour light/dark cycle and constant room temperature. Standard rodent pellets and tap water were provided ad libitum. The animal handling protocols in this study adhered to the ethical guidelines by Universiti Kebangsaan Malaysia Animal Ethics Committee (UKMAEC) (No. Approval: FSK/2018/SATIRAH/23-JAN./891-JAN.-2018-DEC.-2021).

All rats were randomly divided into five groups, namely, A: saline + DMSO (control), B: saline + SNA 209 (SNA), C: isoprenaline + DMSO (ISO), D: isoprenaline + SNA 209 (ISO + SNA), E: isoprenaline + digoxin (ISO + DIG). The ISO was dissolved in normal saline. Saline or isoprenaline was administered 10 mg/kg subcutaneously daily for 14 days to induce heart failure [8]. 17βH-neriifolin (SNA209) was given orally 5 mg/kg next 14 days in the saline group (B) and ISO group (D). Digoxin was given orally 10 mg/kg for the next 14 days in group E.

### 2.3. Blood Pressure and ECG Measurement

Throughout the experiment, blood pressure (BP) and heart rate were measured on days 0, 14, and 28, using the non-invasive tail-cuff method (CODATM non-invasive blood pressure system, Kent Scientific, Torrington, CT, USA) [8]. ECG measurement was carried out in anesthetized rats and placed in a supine position from all the groups on the 28th day. After 30 min of complete anesthesia induction, acupuncture needle electrodes were inserted subcutaneously according to the lead II (right foreleg, left rear leg, and left foreleg) as per the ECG scheme. ECG was recorded for 1 min every 5 min in anesthetized rats by using a PowerLab connected with BioAmp and analyzed by the LabChart 7 software (ADInstrument, Bella Vista, New South Wales, Australia). Each channel was amplified and sampled at a rate of 2 KHz and 5 mV range of a high-pass filter setting of 1 Hz. Alterations in ECG patterns (ST depression or elevation, RR, QRS, and QT interval) in experimental and normal rats were recorded and analyzed [9].

### 2.4. Langendorff Analysis

For the measurement of cardiac function, rat hearts were isolated and perfused ex vivo using the Langendorff method. On day 29, the rats were anesthetized and sacrificed. Blood was collected for further analysis. Before thoracic surgery, rats were anesthetized with ketamine/xylazine (1 mL/kg, i.v.) and administered with heparin (500 IU, i.p.) to prevent blood clotting. Upon loss of pedal reflex activity, the heart was rapidly excised and placed in an ice-cold perfusion buffer before being subjected to retrograde perfusion via the aortic cannula of the Langendorff isolated heart apparatus. Each heart was perfused retrogradely in constant pressure mode (~80 mmHg) with Krebs-Henseleit buffer (in mM: NaCl 118.0; KCl 4.7; MgSO_4_ 1.2; NaHCO_3_ 25.0; KH_2_PO_4_ 1.2; CaCl_2_ 2.5; glucose 11.0) with pH 7.4 and was continuously aerated with 95% O_2_ and 5% CO_2_ at 37 °C. A water-filled latex balloon connected to a pressure transducer (MLT844, ADInstrument) was inserted through the mitral valve into the left ventricle to allow isovolumic contraction and measurement of intraventricular pressure changes. After 20 min of stabilization, cardiac function parameters, such as left ventricular developed pressure (LVDP), left ventricular (LV) maximum and minimum rate of pressure changes (LVdP/dtmax and LVdP/dtmin), as well as the time constant of isovolumic relaxation (Tau), were assessed using a PowerLab data acquisition system. The rate of coronary flow was also recorded by measuring the amount of perfusate flow out from the coronary in one minute, whereas the rate of pressure product (RPP) was formulated from LVDP × HR. All data were acquired using chart software (LabChart 7.0, ADInstrument) as previously described [10].

### 2.5. Analysis of Heart Oxidative Stress Markers

Heart tissue samples were homogenized in chilled Tris-HCl, pH 7.5 and then centrifuged at 12,000 rpm for 10 min at 4 °C. The supernatant was collected and used for measurement of reduced glutathione (GSH) level and thiobarbituric acid-reactive substance (TBARS) level. All markers were normalized to total protein in heart homogenate, which was predetermined using the Bradford assay.

TBARS assay was used to detect malondialdehyde, the major lipid peroxidation product in the homogenate. Heart homogenate reacted with thiobarbituric acid under an acidic environment and boiled at 100 °C for 1 h. Pink color development was measured after cooling down the mixture at 532 nm wavelength, and the TBARS concentration was expressed as mM/mg protein using a standard curve generated with 1,1,3,3-tetraethoxypropane [9].

GSH level in the heart homogenate was measured using the Ellman assay as previously described. Heart homogenate was mixed in Tris-HCl reaction buffer (pH 8.0) and was reacted with 5,5′-dithiobis-2-nitrobenzoic acid (DTNB) for 15 min in the dark. Color development after 15 min was measured at 412 nm wavelength, and the result was expressed as mmol/mg protein using a standard curve generated with GSH [9].

### 2.6. Cardiac Injury Marker

Serum level of troponin T and NT-pro-B-type natriuretic peptide (NT-ProBNP) was quantitatively measured using a commercial ELISA kit (Elabscience Biotechnology Co. Ltd., Wuhan, Hubei, China) following the manufacturer’s protocol. The absorbance of the reaction mixture was read spectrophotometrically at 450 nm [8].

### 2.7. Molecular Analyses

The expression of proteins of interest was analyzed via Western blotting. Protein samples of 10 µL were loaded. After treatment day, heart tissues were lysed in RIPA buffer (Sigma-Aldrich, St. Louis, MO, USA) containing a protease inhibitor cocktail (Roche, Basel, Switzerland) and phosphatase inhibitor (Roche, Basel, Switzerland). The protein content in supernatants obtained after 30 min of centrifugation at 14,000 rpm (4 °C) was quantified by using a Bradford assay (Sigma Aldrich, St. Louis, MO, USA). Proteins (30 µg) were applied to sodium dodecyl sulfate-polyacrylamide gel electrophoresis (10–12%) and then moved to polyvinylidene fluoride (PVDF) membranes (Bio-Rad Laboratories, Hercules, CA, USA). Membrane blocking was performed for 1 h using 5% *w/v* bovine serum albumin or 5% skimmed milk powder in 1% Tris-buffered saline/1% Tween 20. Next, the PVDF membranes were incubated overnight at 4 °C with primary antibodies against (Na^+^/K^+^-ATPase α1, 1:5000) (Santa Cruz Biotechnology, Inc., Santa Cruz, CA, USA), (SERCA2a, 1:1000) (Santa Cruz Biotechnology, Inc., Santa Cruz, CA, USA), (NCX, 1:800) (Santa Cruz Biotechnology, Inc., Santa Cruz, CA, USA) and (β-actin, 1:1000) (Santa Cruz Biotechnology, Inc., Santa Cruz, CA, USA). After thorough washing, the membranes were incubated with horseradish peroxidase-conjugated IgG secondary antibody (1:3000; anti-mouse 7076P2; anti-rabbit 7074S; Cell Signalling Technology, Danvers, MA, USA) for 1 h at room temperature, then at room temperature for 2 h, followed by incubation with secondary antibodies (1:1000) for 1 h. The bands were visualized via enhanced chemiluminescence developing solution (Bio-Rad, Hercules, CA, USA) using a gel documentation system (AI600RGB, GE Healthcare Japan Corporation, Hino, Tokyo, Japan). The bands were then quantified using ImageJ version 1.52v software (US National Institutes of Health, Bethesda, MD, USA). β-actin was used as the control for semiquantitative analysis. The comparative expression level of Na^+^/K^+^-ATPase α1, SERCA2a, and NCX was presented as the IA ratio of Na^+^/K^+^-ATPase α1/β-actin, SERCA2a/β-actin, and NCX/β-actin [11,12].

### 2.8. Histology Studies

Removed hearts were immediately fixed in buffered formalin (10%) for histopathological examination. The left ventricular mass from the apex to the base was sectioned and embedded (with paraffin) after dehydration with alcohol and xylene clearance. Staining of 5 μm-thick histological sections was carried out with eosin and hematoxylin and were examined under the light microscope. Picrosirius red staining was used to evaluate the organization of collagen fibers in heart tissues and viewed under light microscopy as previously described [13]. Quantitative measurements of cardiomyocyte cross-sectional area (circumferential length of myocytes) and percentage of collagen deposition were calculated using ImageJ version 1.52v software (US National Institutes of Health, Bethesda, MD, USA) [13].

### 2.9. Statistical Analysis

Data analyses were performed using GraphPad Prism 8. All data were expressed as mean ± SEM. Shapiro–Wilk test was performed for the normality. Statistical analysis was performed using one-way and repeated measures two-way analysis of variance (ANOVA). The Tukey post hoc test was performed, and statistical significance was set at *p* < 0.05.

## 3. Results

### 3.1. Heart and Left Ventricle Weight Were Not Altered by SNA Treatment

The mortality rate in this study was 5%, which was influenced by isoprenaline injection due to heart failure complications. For this reason, the sample size used for the study was altered from eight to seven animals per group. After the experiments, heart and left ventricle weight were found to significantly increase in ISO groups compared to control (*p* < 0.05). The treatment of SNA and digoxin can reduce the weight of the heart and left ventricle, but it is not statistically significant (Table 1).

### 3.2. Cellular Injury Markers Were Reduced After the SNA Treatment

Following the treatment regimen, serum concentrations of troponin T and N-terminal pro-B-type natriuretic peptide (NT-proBNP) were quantified. As anticipated, the isoprenaline (ISO) group exhibited a significant elevation in both biomarkers—specifically, an increase in NT-proBNP (395.90 ± 26.11 Pg/mL) and troponin T (58.51 ± 2.61 Pg/mL) when compared to the control group. Conversely, the group treated with an SNA demonstrated a statistically significant reduction (*p* < 0.05) in both NT-proBNP (165.80 ± 42.60 Pg/mL) and troponin T (30.56 ± 4.39 Pg/mL). These findings suggest that the SNA treatment possesses cardioprotective and antioxidant properties, which mitigated the cardiac stress-induced damage observed in the ISO group. The detailed biomarker data are presented in Table 1.

### 3.3. SNA Treatment Is Able to Improve Electrocardiogram (ECG) and Blood Pressure Parameters

The present investigation of the electrocardiogram (ECG) which was measured on the 28th day revealed a substantial (*p* ≤ 0.05) prolongation R-R interval (ms) (0.54 ± 0.01), QRS complex (ms) (0.026 ± 0.001), QT interval (ms) (0.035 ± 0.001) in the ISO group relative to the control group, indicating the presence of cardiac arrhythmia. As demonstrated in Figure 1 and Table 2, however, the R-R interval (ms) in ISO + SNA 5.0 (0.38 ± 0.01) indicates a significant (*p* ≤ 0.05) decrease as compared to ISO (0.54 ± 0.01). QRS complexes and QT interval were significantly (*p* ≤ 0.05) shortened in ISO + SNA 5.0 (0.029 ± 0.001) and (0.028 ± 0.001), respectively. 

SBP, DBP, MAP, and HR were measured at baseline, on the 14th and 28th day. As shown in Table 3, blood pressure measurements were significantly (*p* < 0.05) increased from baseline to day 14 in the ISO and all treatment groups compared to the control group. However, the SNA treatment was able to improve blood pressure measurements, resulting in a significant reduction over the next 14 days, until the 28th day. A significant reduction (*p* < 0.05) compared to the control group in heart rate was also observed in the first 14 days in ISO and treatment groups, but the SNA treatment was able to increase the HR to the normal range in the treatment group. 

### 3.4. SNA Enhances the Contractility in Isolated Perfused Rat Hearts

Cardiac mechanical function in isolated perfused rat hearts was measured by the Langendorff method. Left ventricle developed pressure (LVDP) was significantly (*p* < 0.05) reduced in the ISO group compared to the control. However, SNA treatment was able to improve the LVDP, thus improving the contractility of a failed heart. This is further supported significantly by all other Langendorff measurements such as left ventricular (LV) maximum and minimum rate of pressure changes (LVdP/dtmax and LVdP/dtmin) as well as the time constant of isovolumic relaxation (Tau) (Figure 2A–E).

### 3.5. SNA Is Able to Mitigate Oxidative Stress

Table 1 presents the TBARS levels in the heart homogenate for all experimental groups. After 14 days of administration, the TBARS level in the ISO group was significantly higher compared to the vehicle controls (*p* < 0.05). However, co-administration of SNA and digoxin significantly prevented this rise in TBARS levels when compared to the ISO group (both *p* < 0.05). A concomitant reduction in GSH levels accompanied the elevated TBARS levels in ISO-administered rats. While both SNA and digoxin similarly increased GSH levels in the heart homogenate compared to the ISO group, this effect was not statistically significant (both *p* > 0.05).

### 3.6. SNA Improves Cardiac Hypertrophy and Fibrosis in Rats

As shown in Figure 3A–E, the results demonstrated that the increased cardiomyocyte size was significant, as disclosed by H&E staining in the ISO group compared to the control group. This is further supported by the quantitative measurements of cardiomyocyte cross-sectional area (circumferential length of myocytes) that ISO had significantly higher area compared to the control group, and SNA treatment was able to improve the hypertrophic condition (Figure 3F). In addition, SNA treatment decreased the ISO-induced accumulation of fibrosis collagen in the myocardial interstitial space (Figure 4A–E). This is also further supported by the measurement of collagen deposition (%) in Picrosirius red-stained left ventricle sections, which revealed a significantly higher percentage of collagen deposition in the ISO group compared to the control (Figure 4F). However, SNA was able to reduce the fibrosis condition significantly in the treatment group compared to the ISO group.

### 3.7. SNA Increases the Protein Expression Related to Contractility in Rats

Intracellular calcium homeostasis and contractility mainly depend upon α1 Na^+^K^+^ ATPase, SERCA-2a, and NCX. Therefore, the expression of these proteins was analyzed in the right ventricle of HF model. Total α1 Na^+^K^+^ ATPase, SERCA-2a, and NCX protein levels in the right ventricle of the SNA and digoxin treatment group were significantly higher than the HF group, which may explain the differences in performance between the groups (Figure 5) and the supplementary blot are listed Appendix A. The SNA treatment was able to increase the cardiac total α1 Na^+^K^+^ ATPase and SERCA-2a expression by enhancing SR Ca^2+^ uptake resulting in several beneficial effects such as improved systolic and diastolic functions, increased cardiac contractility, and survival and resistance to HF during prolonged pressure overload.

## 4. Discussion

Heart failure (HF) is a complex clinical illness that arises from anatomical and blood-pumping system deficiencies. It has a complex etiology and manifestation. The heart often experiences complex remodeling as the body adapts to various risk factors, leading to distinct anatomical and physical changes. It may eventually lead to systolic dysfunction, arrhythmias, and ventricular dilatation [12,14]. The hallmark of cardiac remodeling is cardiomyocyte hypertrophy. There are also progressive alterations in the management of calcium (Ca^2+^) and contractile function [15]. The isoprenaline heart failure animal model, which induces cardiac damage and remodeling via sustained β-adrenergic receptor stimulation, is a highly reproducible and widely used model for this specific purpose [8,16,17]. While it may not fully capture the genetic diversity or complex comorbidities of human heart failure, it effectively produces key features, such as myocardial hypertrophy, cardiomyocyte necrosis, and fibrosis, which are crucial endpoints in mimicking clinical heart failure features [17,18,19,20,21]. The choice of a 14-day treatment period in our study is based on the pharmacokinetics of the drugs used. Digoxin, a common cardiac glycoside, has a half-life of approximately 36 to 48 h in humans with normal kidney function. Since it takes about five half-lives for a drug to reach a steady-state concentration, a treatment period of around seven to ten days is generally needed. A 14-day period provides a safe buffer, ensuring the drug has reached its full therapeutic potential, and the researchers can observe its sustained effects on cardiac function and heart failure symptoms [3,22,23]. Our major novel findings from 14 days of SNA 209 treatment were that SNA 209 was able to reduce the cellular injury markers, normalize the ECG parameters, increase the cardiac output and function as well as restore the cardiac calcium handling protein.

In the current work, the cardio-protective effect of SNA 209 treatment on the ISO-induced model was investigated via several parameters. First and foremost is cellular injury markers troponin T and N-terminal prohormone of B-type natriuretic peptide (NT-ProBNP) were measured, and SNA 209 was able to reduce the cellular injury markers after the treatment. Isoprenaline (ISO), a synthetic catecholamine, is commonly used to induce heart failure in experimental models. Isoprenaline exposure mimics the effects of chronic sympathetic stimulation, leading to excessive stress by releasing reactive oxidative stress (ROS) in heart muscle cells. This stress can cause damage to the sarcolemmal membrane (cell membrane) and myofibrils (contractile units) within cardiomyocytes (heart muscle cells). Damaged cardiomyocytes leak troponin T, a protein specific to the sarcomere, into the bloodstream. This leakage elevates blood troponin T levels, indicating myocardial injury. As reported in the previous study, in response to a failing heart, the body releases hormones like NT-proBNP to counteract stress and maintain blood pressure. NT-proBNP is produced by the ventricles of the heart and acts as a vasodilator and diuretic. Increased NT-proBNP is a hallmark of heart failure and indicates the heart’s attempt to compensate for its weakened pumping ability [7,24,25]. Studies suggest that 17βH-neriifolin (SNA 209) exhibits antioxidant properties. It can scavenge free radicals, reactive oxygen species (ROS), and other harmful molecules produced during cellular stress or injury. By neutralizing these reactive molecules, 17βH-neriifolin can help protect cellular components like proteins, lipids, and DNA from oxidative damage, a significant contributor to cell injury [26,27].

Oxidative stress is often linked to cardiac injury. The auto-oxidation of isoprenaline produces quinones, which react with oxygen to form hydrogen peroxide and superoxide anions. Together, they deplete endogenous antioxidant enzymes and cause massive oxidative stress. This balance is thus upset by the administration of isoprenaline, which raises the need for antioxidant enzymes. In addition to causing damage to cardiomyocyte cells, upregulation of ROS production and the concurrent decrease in endogenous antioxidants also fuel the fibrosis and inflammation associated with heart diseases [21]. In this investigation, the ISO group’s TBARS level—a measure of lipid peroxidation—rose noticeably higher than that of the vehicle controls. Following 14 days of isoprenaline administration, there was a concurrent decrease in the heart’s GSH level, which together suggested an increased level of oxidative stress. This finding was consistent with a number of earlier investigations [21,28]. Similar to digoxin, SNA administration dramatically reduced TBARS levels significantly when compared to the ISO group. Increased TBARS may result from increased ROS production, which may encourage lipid membrane peroxidation in the heart. One of the most significant non-enzymatic antioxidants, GSH keeps the redox state of different cells stable and shields membrane lipids from oxidation. Numerous reports suggested that GSH depletion is linked to tissue damage brought on by different stimuli [28,29]. Long-term ISO administration raises ROS and depletes GSH in the heart and aorta, as we and others have previously demonstrated [8,29]. However, SNA treatment improves the GSH level compared to the ISO group. These changes in TBARS and GSH levels proved that SNA has antioxidant properties.

Electrocardiographic (ECG) abnormalities serve as key indicators for diagnosing heart failure. In the present study, isoproterenol (ISO) administration in rats led to several significant ECG changes, including ST-segment elevation and a prolongation of the R-R interval, QRS complex, and QT interval. These findings suggest the presence of cardiac arrhythmia. Such abnormalities in ISO-induced heart failure are likely caused by the generation of free radicals and subsequent oxidative stress, which compromise cell membrane function and disrupt cardiac conduction. A prolonged QRS interval specifically points to ventricular dysfunction [21,30]. These results align with a study by Yousry et al. (2021) [21], which also reported an increase in ST-segment elevation. They attributed this finding to disturbances in the heart’s conduction system caused by chronic overstimulation of the β-adrenergic receptor. The treatment group successfully normalized the elevated ST-segment, R-R interval, QRS complex, and QT interval. This suggests that the compound 17βH-neriifolin possesses cell membrane-stabilizing potential, likely due to its potent antioxidant properties.

The current study found that systolic blood pressure (SBP) and diastolic blood pressure (DBP) were significantly (*p* < 0.05) higher in isoprenaline (ISO)-induced rats. This finding is consistent with a previous study, which suggested that repeated exposure to low-dose ISO could activate the sympathetic nervous system via β1-adrenoceptor stimulation, leading to an increase in blood pressure [31]. In contrast to the control group, the heart rate of the ISO-induced rats was notably lower. This decrease, along with the other hemodynamic changes, may indicate a general malfunction of cardiac contractility and relaxation [32]. Interestingly, 17βH-neriifolin functions as a cardiac glycoside. It works by inhibiting the efflux of sodium from cardiac cell membranes, thereby increasing intracellular calcium concentration [33]. This calcium accumulation subsequently enhances the force of myocardial contraction, resulting in a greater stroke volume and, consequently, an increase in cardiac output. This increase in cardiac output can lead to a compensatory decrease in heart rate while still effectively reducing blood pressure compared to the ISO group.

The progression of cardiac hypertrophy can lead to the loss of ventricular function, which is characterized by a decrease in key cardiac parameters such as left ventricular developed pressure (LVDP), tau, and the rates of contraction and relaxation, as is commonly observed in heart failure. Studies have demonstrated that hearts isolated from rats with ISO-induced cardiac hypertrophy exhibit a reduction in both the amplitude of the intracellular calcium transient and the expression of sarcoplasmic reticulum Ca^2+^-ATPase (SERCA). This reduction diminishes the calcium load in the sarcoplasmic reticulum, thereby contributing to reduced cardiac contractility [16,34,35]. In the current study, left ventricular dysfunction induced by heart failure was evident in Langendorff-perfused rat hearts. This was shown by a prolonged relaxation time and a reduced rate of ventricular relaxation, both of which indicate impaired left ventricular relaxation. The further reduction in LVDP, rate of ventricular contraction, and coronary flow rate suggests that ventricular contraction was also impaired due to the chamber’s stiffening caused by isoprenaline. However, the administration of SNA 209 as a cardiac glycoside reversed the trend of these cardiac mechanical parameters, suggesting that the progression of cardiac dysfunction was alleviated. This finding is supported by a study from Ramalingam et al. (2020), which found that SNA 209 (17βH-neriifolin) enhances left ventricular contractility in both normal and failing hearts ex vivo [36].

Administration of the synthetic catecholamine isoprenaline (ISO), a beta-receptor agonist, is a well-established animal model for inducing myocardial damage, cardiac hypertrophy, left ventricular (LV) dysfunction, and heart failure [16,19,34]. In this study, rats in the ISO group showed significant cardiomyocyte enlargement after 28 days, as observed in hematoxylin and eosin-stained heart sections. This was statistically significant compared to the control group, as confirmed by quantitative analysis of cardiomyocyte area. The chronic, low-dose ISO exposure in rats resulted in myocardial damage and hypertrophy, which aligns with observations from previous studies. Furthermore, our findings indicate that SNA 209 treatment attenuated this beta-receptor agonist-induced cardiac hypertrophy and also reduced the hypertrophic response in cardiomyocytes in an ex vivo setting. Myocardial remodeling leads to cardiac fibrosis, a process characterized by the deposition of interstitial collagen, which can impair cardiac function and contribute to the development of heart failure [17,18,19]. The ISO group exhibited a marked increase in collagen deposition, a finding supported by the quantification of heart sections, which showed a significantly higher percentage of collagen compared to the control group. Treatment with SNA 209 not only attenuated the differentiation of fibroblasts into myofibroblasts but also improved the symptoms of myocardial fibrosis. The antioxidant properties of SNA 209 are proposed to be responsible for improving both hypertrophy and collagen deposition by mitigating the oxidative stress caused by isoprenaline.

In animal models of heart failure, elevated sodium (Na^+^) levels have been shown to increase the activity of Na^+^/K^+^ ATPase, SERCA2a, and NCX, which can lead to deteriorated diastolic function, arrhythmia, compromised cell metabolism, and oxidative stress [37,38]. In this study, we observed a significant reduction in the expression of SERCA2a, α1 Na^+^/K^+^ ATPase, and NCX in the isoprenaline (ISO)-induced group compared to the control group. This reduction in calcium-handling protein expression may be attributed to elevated lipid peroxidation [34,38,39,40]. 17βH-neriifolin (SNA209), as a cardiac glycoside, is a reversible inhibitor of the Na+-K+-ATPase. It acts similarly to ouabain or digoxin by binding to the α-subunit of the enzyme, thereby inhibiting its function. This binding is reversible, meaning the inhibitor can detach from the enzyme, potentially allowing the enzyme to regain its function [6]. As a cardiac glycoside, SNA 209 was able to restore these calcium-handling proteins in the failing hearts, which helps strengthen abnormal contractions through Ca^2+^ regulation [37,39,40]. The cardioprotective effect of SNA 209 on cardiac remodeling in a heart failure rat model was comparable to that of digoxin. Digoxin is known to improve heart failure through multiple mechanisms, including its effects on cardiac contractility, electrical activity, and the neurohormonal system [3,29,41]. Our results indicate that both digoxin and SNA 209 treatment not only restored protein expression but also did not worsen the disease. The ability of both compounds to restore Na^+^/K^+^ ATPase protein levels in the ISO-treated animals suggests a potential long-term, anti-remodeling effect that extends beyond their immediate pump-inhibitory function. This effect may involve the modulation of cellular signaling pathways that influence gene expression, which is a known characteristic of cardiac glycosides [29,41]. Collectively, these findings support the conclusion that SNA 209 could alleviate the damaging effects of myocardial hypertrophy and fibrosis and play a cardioprotective role in the failing heart.

## 5. Conclusions

In summary, this study demonstrates that low-dose isoproterenol exposure induces significant cardiac damage, including changes in heart function, tissue injury, and the development of hypertrophy and fibrosis. We show that the cardiac glycoside 17βH-neriifolin (SNA 209) effectively reverses these effects by restoring heart function and repairing tissue damage, likely through its action on calcium handling proteins. These promising results suggest that SNA 209 could be a valuable new treatment for heart failure.

## 6. Future Studies

Future research should focus on two key areas to build on these findings. First, a more detailed investigation into the phosphorylation of phospholamban (PLN) is needed to fully understand its role in SNA 209’s effects. Second, we recommend further studies to confirm the direct link between SNA 209 and the Nrf2/ARE pathway, which would clarify its antioxidant mechanism. These studies will provide critical mechanistic insights, paving the way for the clinical development of SNA 209 as a therapeutic agent.

## Figures and Tables

**Figure 1 biomedicines-13-02115-f001:**
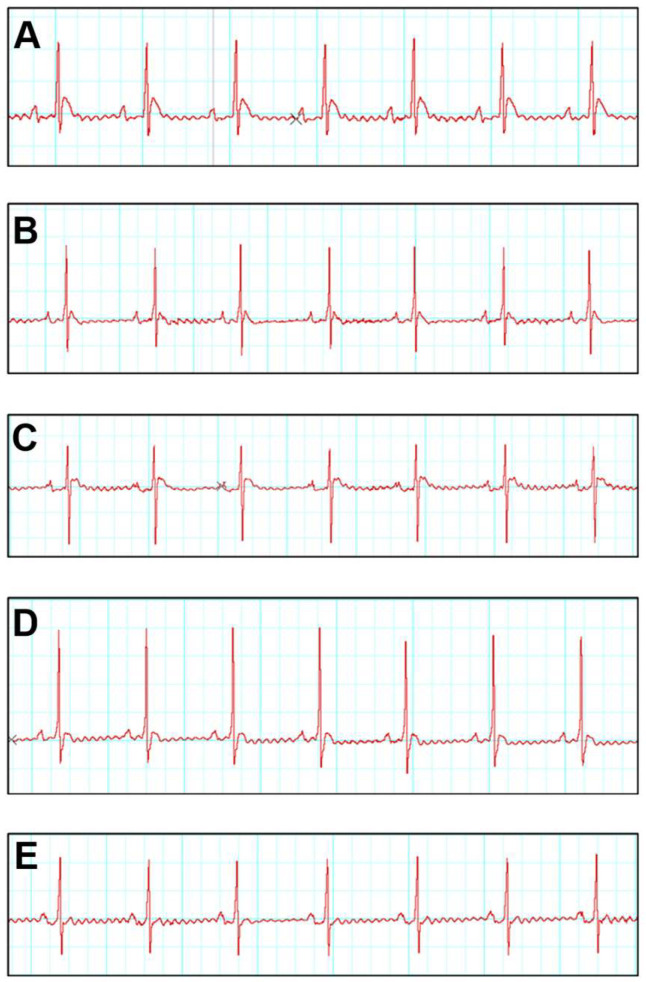
Effects of ISO and SNA treatment on the electrocardiogram (ECG). Representative ECG tracing. (**A**) Control, (**B**) SNA 5.0, (**C**) ISO, (**D**) ISO + SNA 5.0, and (**E**) ISO + Dig. (*n* = 7 or 8 per group). ISO; isoprenaline. Dig; digoxin.

**Figure 2 biomedicines-13-02115-f002:**
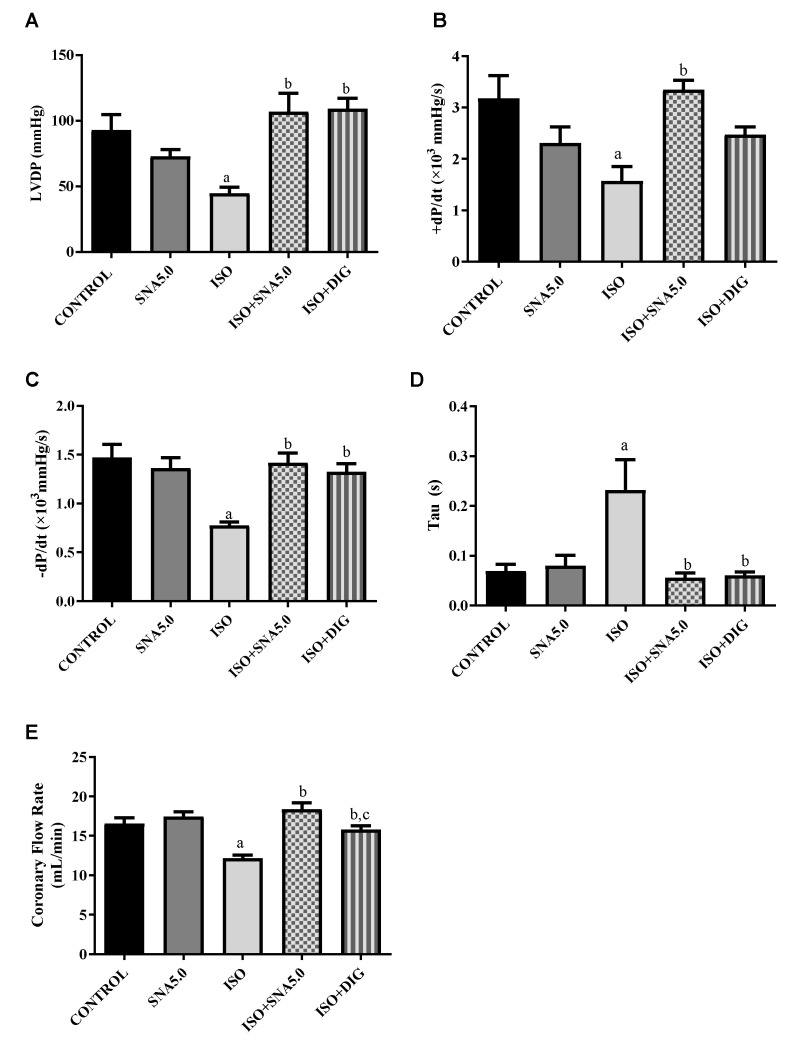
Analysis of cardiac mechanical function in isolated perfused rat hearts. (**A**) LVDP, (**B**) LVdP/dtmax, (**C**) LVdP/dtmin, (**D**) Tau, and (**E**) coronary flow. Values are expressed as the mean ± SEM (*n* = 7 or 8 per group). ^a^ *p* < 0.05 in relative to control group, ^b^ *p* < 0.05 in relative to ISO group and ^c^ *p* < 0.05 in relative to ISO + SNA5.0 group. ISO; isoprenaline. DIG; digoxin.

**Figure 3 biomedicines-13-02115-f003:**
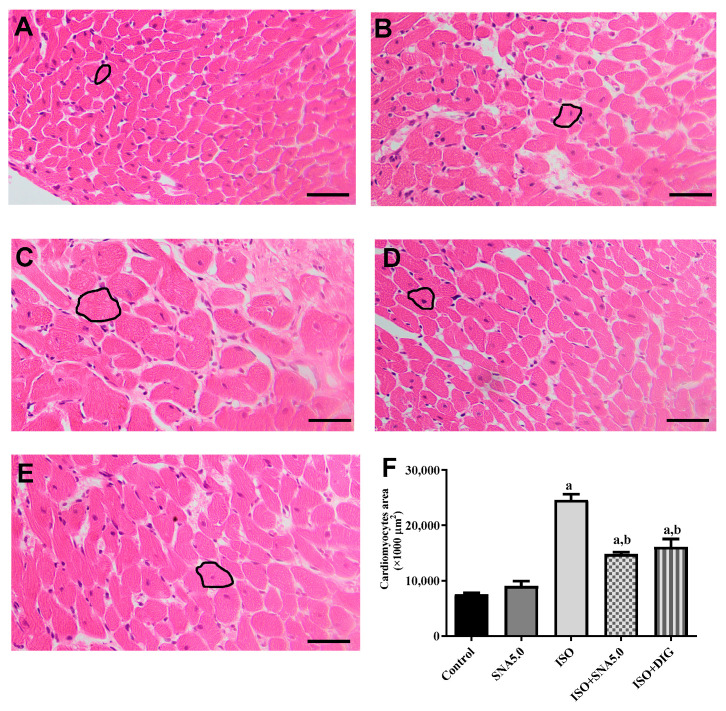
Representative × images of H&E-stained left ventricle sections; 40× magnification; scale bar 50 μm. Representative cardiomyocytes circled in black show the cardiomyocyte area measured. ((**A**) control, (**B**) SNA5.0, (**C**) ISO, (**D**) ISO + SNA5.0, and (**E**) ISO + DIG). (**F**) The measurements of cardiomyocyte cross-sectional areas in H&E-stained left ventricle sections. Values are expressed as mean ± S.E.M, where *n* = 5–7. ^a^ *p* < 0.05 in relative to control group, ^b^ *p* < 0.05 in relative to the ISO group.

**Figure 4 biomedicines-13-02115-f004:**
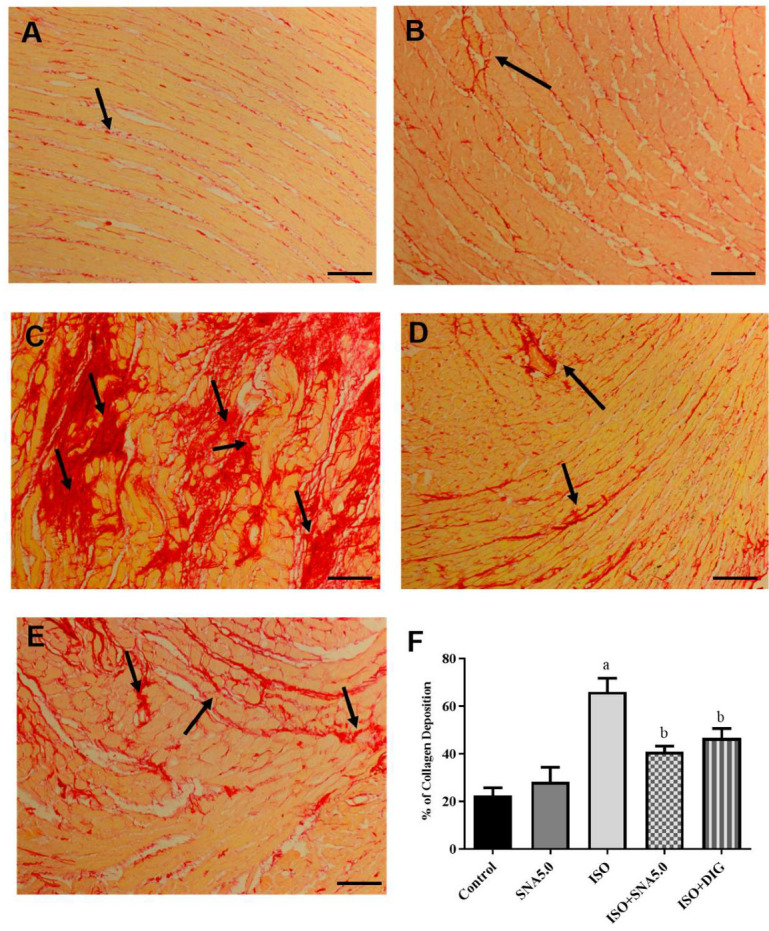
Representative images of Picrosirius red-stained left ventricle sections; 100× magnification; scale bar 50 μm. Arrows show deposition of collagen in red stain. ((**A**) control, (**B**) SNA5.0, (**C**) ISO, (**D**) ISO + SNA5.0, and (**E**) ISO + DIG). (**F**) The measurement of collagen deposition (%) in Picrosirius red-stained left ventricle sections. Values are expressed as mean ± S.E.M, where *n* = 8. ^a^ *p* < 0.05 in relative to control group, ^b^ *p* < 0.05 in relative to ISO group.

**Figure 5 biomedicines-13-02115-f005:**
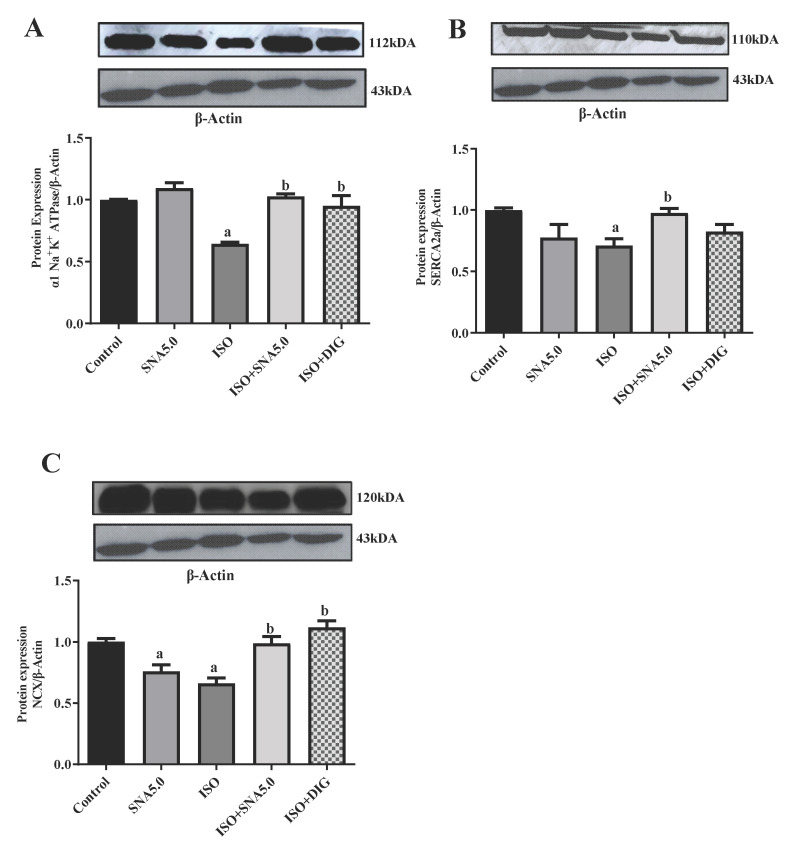
Representative Western blots and quantification of protein expression levels normalized to beta-actin (β actin) and relative to control (*n* = 5–7/group). (**A**) α1 Na^+^K^+^ ATPase, (**B**) SERCA2a, (**C**) NCX. Statistical analysis: two-way ANOVA with Tukey post hoc test was performed, and statistical significance was set at *p* < 0.05. Values are expressed as mean ± S.E.M, ^a^ *p* < 0.05 in relative to control group, ^b^ *p* < 0.05 in relative to ISO group.

**Table 1 biomedicines-13-02115-t001:** The effects of 17βH-neriifolin treatment on rats’ body weight, heart weight, cardiac injury markers, TBARS levels, and GSH level at the end of the experiment.

Group	Heart Weight (HW/g)	HW/TL (g/cm)	LV/TL (g/cm)	Troponin T (Pg/mL)	NT-ProBNP (Pg/mL)	TBARS (nmoL/mg Protein)	GSH (nmoL/mg Protein)
CONTROL	0.85 ± 0.04	0.24 ± 0.01	0.16 ± 0.01	40.49 ± 2.74	157.00 ± 29.08	3.653 ± 0.61	0.655 ± 0.05
SNA 5.0	0.93 ± 0.04	0.26 ± 0.01	0.15 ± 0.01	28.54 ± 5.11	183.60 ± 9.90	2.456 ± 0.38	0.527 ± 0.07
ISO	1.08 ± 0.10 *	0.30 ± 0.01 *	0.21 ± 0.01 *	58.51 ± 2.61 *	395.90 ± 26.11 *	17.66 ± 7.13 *	0.381 ± 0.03 *
ISO + SNA 5.0	1.03 ± 0.04	0.29 ± 0.01	0.21 ± 0.01	30.56 ± 4.39 **	165.80 ± 42.60 **	2.377 ± 0.64 **	0.428 ± 0.04 *
ISO + DIG	1.05 ± 0.04	0.29 ± 0.01	0.20 ± 0.01	57.03 ± 4.84	227.40 ± 16.26 **	2.765 ± 0.22 **	0.449 ± 0.02 *

Values are presented as mean ± SEM for *n* = 8 per group, * *p* < 0.05 in relative to control group, ** *p* < 0.05 in relative to ISO group. HW: heart weight, TL: tibial length, LV: left ventricle, TBARS: thiobarbituric acid-reactive substance, GSH: glutathione.

**Table 2 biomedicines-13-02115-t002:** The effects of 17βH-neriifolin treatment on rats’ electrocardiogram (ECG) parameters at the end of the experiment.

Group	ST Elevation (mV)	RR (s)	QRS-Complex (ms)	QT-Interval (ms)
CONTROL	0.14 ± 0.01	0.33 ± 0.01	0.025 ± 0.001	0.026 ± 0.001
SNA 5.0	0.24 ± 0.03	0.33 ± 0.01	0.026 ± 0.001	0.024 ± 0.002
ISO	0.38 ± 0.02 *	0.54 ± 0.01 *	0.039 ± 0.001 *	0.035 ± 0.001 *
ISO + SNA 5.0	0.17 ± 0.01 **	0.38 ± 0.01 **	0.029 ± 0.001 **	0.028 ± 0.001 **
ISO + DIG	0.22 ± 0.02 **	0.37 ± 0.01 **	0.028 ± 0.001 **	0.028 ± 0.001 **

Values are presented as mean ± SEM for *n* = 7–8 rats per group, * *p* < 0.05 in relative to the control group, ** *p* < 0.05 in relative to ISO group.

**Table 3 biomedicines-13-02115-t003:** The effects of 17βH-neriifolin treatment on blood pressure (BP) and heart rate (HR) in the heart failure rat model.

Group	Systolic Blood Pressure (SBP) (mmHg)			Diastolic Blood Pressure (DBP) (mmHg)			Mean Arterial Pressure (MAP) (mmHg)			Heart Rate (bpm)		
	0th	14th	28th	0th	14th	28th	0th	14th	28th	0th	14th	28th
Control	129.63 ± 6.87	126.42 ± 1.68	128.09 ± 5.05	88.67 ± 6.23	86.07 ± 2.93	85.52 ± 2.83	102.32 ± 6.44	99.52 ± 2.28	99.71 ± 3.57	390.83 ± 24.42	370.79 ± 27.24	359.17 ± 48.45
SNA 5.0	129.88 ± 2.83	130.51 ± 9.93	129.22 ± 13.13 ^a^	88.12 ± 3.88	87.50 ± 5.60	89.23 ± 10.57 ^a^	102.02 ± 3.23	101.68 ± 7.20	102.63 ± 10.79 ^a^	374.02 ± 29.45	349.67 ± 53.13	329.54 ± 60.31 ^a^
ISO	127.29 ± 5.81	139.07 ± 14.33 ^a^	140.28 ± 7.01 ^a^	86.00 ± 5.10	96.14 ± 4.06 ^a^	103.11 ± 10.09 ^a^	99.96 ± 5.21	111.39 ± 6.35 ^a^	114.47 ± 10.01 ^a^	380.61 ± 23.72	283.78 ± 65.01 ^a^	267.29 ± 42.02 ^a^
ISO + SNA 5.0	127.40 ± 3.45	140.63 ± 3.17	127.52 ± 4.15 ^b^	85.46 ± 2.21 ^b^	100.30 ± 8.15	85.61 ± 4.39 ^b^	99.44 ± 2.83	113.74 ± 8.50	99.58 ± 2.42 ^b^	409.37 ± 24.82	262.81 ± 37.33 ^b^	333.15 ± 33.10 ^b^
ISO + DIG	128.50 ± 4.22	138.31 ± 7.94	131.3 ±1 0.77 ^b^	86.64 ± 4.17	97.39 ± 10.44	88.19 ± 6.08 ^b^	100.60 ± 3.15	111.03 ± 10.80	102.56 ± 6.01 ^b^	391.36 ± 19.70	279.60 ± 41.55 ^b^	333.08 ± 38.84 ^b^

Values are presented as mean ± SEM for *n* = 8 per group, ^a^ *p* < 0.05 in relative to the control group, ^b^ *p* < 0.05 in relative to the ISO group. ISO; isoprenaline. DIG; digoxin.

## Data Availability

The original contributions presented in this study are included in the article. Further inquiries can be directed to the corresponding authors.

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
