# Peer review of "17βH-Neriifolin Improves Cardiac Remodeling Through Modulation of Calcium Handling Proteins in the Heart Failure Rat Model"

_biomedicines, 2025, doi:10.3390/biomedicines13092115_

Round 1

Reviewer 1 Report

Comments and Suggestions for Authors

1. Clarify Data Presentation:

- Clearly label all figure panels. Include raw Western blot images.

- Fix the formatting of Table 1 and define all abbreviations (e.g., "HW/TL," "LV/TL").

2. Expand Experimental Design:

- Include multiple doses of SNA209 to establish efficacy and safety margins.

- Assess arrhythmia risk through detailed ECG analysis (e.g., QT interval, ectopic beats).

 - Extend the treatment duration to evaluate long-term effects.

3. Deepen Mechanistic Insights:

- Investigate the phosphorylation of phospholamban, a key regulator of SERCA2a.

- Measure intracellular calcium transients in cardiomyocytes.

- Explore the Nrf2/ARE pathway to link antioxidant effects to functional improvements.

4. Strengthen Discussion:- Compare the therapeutic index of SNA209 directly to that of digoxin using toxicity data.

- Address the limitations of the ISO model in mimicking human HF.

Reviewer 2 Report

Comments and Suggestions for Authors

The present study reports the cardioprotective effects of SNA against isoproterenol-induced heart failure in rats. The findings are novel and acceptable for publication. However, there are potential errors and issues to be addressed by authors. Please refer to the following.

  1. Line 27: rate, it should be heart rate. Please correct it.
  2. Line 19-20 indicates SNA inhibited the Na+-K+-ATPase, whereas 32-33 indicates SNA increases Na+-K+-ATPase expression of the α1 subunit. Isn't this statement contraindicated?
  3. Rewrite lines 72-74.
  4. How was the 17βH-neriifolin (SNA209) dose confirmed? Instead of a single dose, authors could have used 3 doses to confirm its dose-dependent and toxic effects. Mention the dose of ISO.
  5. Please confirm whether 17βH-neriifolin (SNA209) inhibits the Na+-K+-ATPase reversibly or irreversibly, which is important in assessing the risk-benefit ratio of cardiac glycosides.
  6. The blood pressure and troponin details in the abstract are missing.
  7. Figure 1D: The significant differences are wrongly indicated. Please check.
  8. Figure 4 labelling must be improved.
  9. It seems that SNA exhibited better protection than digoxin. However, authors have missed to write their choice as a cardioprotectant other than digoxin in the discussion section. Therefore, I suggest improving the discussion.
  10. Please add the representative ECG graphs and the changes in the various ECG parameters.
  11. The cite reference number 4: This article did not identify the SNA role in Na+-K+-ATPase activity. Therefore, add a suitable reference.

Round 2

Reviewer 2 Report

Comments and Suggestions for Authors

Authors have addressed all the issues